# Elite Football Coaches Experiences and Sensemaking about Being Fired: An Interpretative Phenomenological Analysis

**DOI:** 10.3390/ijerph17145196

**Published:** 2020-07-18

**Authors:** Marte Bentzen, Göran Kenttä, Pierre-Nicolas Lemyre

**Affiliations:** 1Department of Teacher Education and Outdoor Studies, The Norwegian School of Sport Sciences, 0806 Oslo, Norway; 2The Swedish School of Sport and Health Sciences, 114 33 Stockholm, Sweden; Goran.Kentta@gih.se; 3The School of Human Kinetics, University of Ottawa, Ottawa, ON K1N 6N5, Canada; 4Department of Sport and Social Sciences, The Norwegian School of Sport Sciences, 0806 Oslo, Norway; nicolasl@nih.no

**Keywords:** job insecurity, stress, performance expectations, dismissals, occupational health

## Abstract

Background: Chronic job insecurity seems to be a prominent feature within elite sport, where coaches work under pressure of dismissals if failing to meet performance expectations of stakeholders. The aim of the current study was to get a deeper understanding of elite football coaches’ experiences of getting fired and how they made sense of that process. Method: A qualitative design using semi-structured interviews was conducted with six elite football coaches who were fired within the same season. Interpretative phenomenological analysis was chosen as framework to analyze the data. Results: The results reflected five emerging themes: Acceptance of having an insecure job, working for an unprofessional organization and management, micro-politics in the organization, unrealistic and changing performance expectation, and emotional responses. Conclusion: All coaches expressed awareness and acceptance regarding the risk of being fired. However, they experienced a lack of transparency and clear feedback regarding the causes of dismissal. This led to negative emotional reactions as the coaches experienced being evaluated by poorly defined expectations and by anonymous stakeholders. Sports organizations as employers should strive to be transparent during dismissal. In addition, job insecurity is a permanent stressor for coaches and should be acknowledged and targeted within coach education.

## 1. Introduction 

Coaching in competitive sports is a profession that has the potential to be rewarding from a wide range of perspectives but also a profession known for its high demands and pressures, especially at the highest level. Coaches typically face a large number of varied demands associated with the working context of elite sports, such as: unrealistic performance demands, management, administration, interpersonal and organizational stressors [1,2]. Moreover, the context is complex, dynamic, unpredictable, highly competitive, and involves extensive amounts of travel days and work hours without boundaries that may lead to work and home life interference. Stress is and will always be an integral part of elite sport, and key features of coaching at this level are unrealistic performance demands and job insecurity. This is especially relevant for bigger team sports such as football, which also involve thousands (sometimes millions) of engaged and passionate fans, media, stakeholders, and sponsors. Indeed, high performance coaching in general is a profession that is vulnerable to mental health problems [3] and in particular football [4,5,6].

### 1.1. Occupational Health and Job Demands

Chronic stress at work will accumulate over time and gradually result in impaired occupational health [1,3]. Health has been described as a concept that comprises both ill-being and well-being [7], whereas reduced occupational health consequently could include both increase in ill-being indicators (e.g., exhaustion, negative emotions, stress-reactions) and/or reduction in well-being indicators (e.g., vitality, motivation, positive emotions). Previous research on reduced occupational health for high-performance coaches has reported a range of ill-being symptoms, such as depressed mood, fatigue, sleep disturbance, lack of vitality, increased anger, and irritability [8,9,10]. Further, coaches reduced occupational health has also shown to be negatively associated with work motivation [11,12], their relation between private life and work life [9,13], and work performance [8,14]. In order to more comprehensively understand how coaches’ occupational health is affected, it is of importance to better understand the mechanisms and characteristics at work that either reduce well-being indicators or enhances ill-being indicators.

The job demands–resources (JD-R) model [15] has widely been used to better understand how job characteristics influence employees, occupational health, and performance. The revised JD-R model describes how job characteristics can be divided into either job resources, which enhances and support occupational well-being and performance (e.g., skill utilization, supervisor support, financial rewards, and career opportunities) [15] and two qualitatively different types of job demands, namely, job challenges and job hindrances [16,17]. Job challenges are described to be stressors at work that can stimulate the employers capacities but at the same time also be energy depleting, as these job characteristics promote and challenge the employers curiosity and competence, making the employer to strive for improvement through problem-solving, which again can lead to achieving work goals [18]. Applied to the context of coaching, these can be stressors that coaches can handle despite required effort, such as optimizing their collaboration and communication with their athletes or increasing their own tactical knowledge. On the other hand, job hindrances are described to be stressors that hinder optimal functioning, which requires a lot of energy without yielding any benefits [17]. Further, stressors that are characterized as job hindrances are those outside the employers’ control and are expected to enhance their negative emotions [17]. In previous occupational research, role ambiguity, job insecurity, constraints, and interpersonal conflicts are examples of stressor identified as job hindrances [18,19]. Considering recent research on coaches and stressors within high performance sport, all of these stressors are highly recognizable [1,2,20].

### 1.2. Job Insecurity

Job insecurity has received a lot of attention and it is acknowledged as one of the predominant stressors employees have to deal with, in the occupational psychology literature [21]. Job insecurity is defined as the perceived discrepancy between the level of employment security employees experience and the level they prefer [22]. Interestingly, job insecurity has received very limited attention in sports science, despite being a highly recognized stressor that could be classified as a job hindrance. Nevertheless, job insecurity and the risk of job termination is acknowledged as an integral part of the coaching profession [23]. The head coach in football has a key role in their club, which means that they are responsible and accountable for sporting failure [24]. Consequently, football coaches work under pressure of getting fired if the team underperforms and fails to meet the expectations of stakeholders, regardless of how this is evaluated [24,25]. In fact, a third of the football clubs in Sweden (10 out of 32 clubs) from the two top leagues fired their coaches last season. Simply noticing that the bottom 6 teams in one of the leagues fired their coaches brings performance issues to attention. Thus, chronic job insecurity has been described as a prominent feature of football managers and coaches’ work environment, both in England and Scandinavia [24,25,26]. 

### 1.3. Research on Dismissals in Sport 

Research has explored football coach dismissals from a variety of perspectives. Typically, from a results-based perspective [25,27,28,29,30,31], with an emphasis on how the dismissal of a coach affects the results of the team. Despite ambiguous findings, the overall and long-term findings indicate that dismissals will not improve team performance, either in matches won or in goals scored. Further, some studies have focused on whether there is an increasing or decreasing trend of dismissals and based on findings discussed potential explanations behind an increasing trend [32,33]. This trend could be a consequence of public justification, since coaches and the team are under constant pressure to meet high standards and high-performance expectations over time [33]. Although this challenge is part of the context for most teams in competitive elite sport, being a coach and trying to manage unrealistic expectations over time could be considered a “mission impossible” [33]. Several researchers have taken a management perspective when exploring dismissals of soccer coaches both from an economic perspective [27] and a managerial perspective [24]. Based on the economic perspective, findings indicate that coaches who work for clubs with higher team wages are more likely to get sacked sooner, compared with coaches working for clubs with lower team wages [27]. Interestingly, research conducted in Danish elite football applied a qualitative and a more comprehensive managerial approach to examine how managers make sense of dismissing coaches [24]. Central findings that emerged from this study were that poor team performance was often a starting point, followed by external pressure (i.e., fans and media), resulting in dismissals because clubs want to live up to expectations as “clubs of action” from a managerial stand point [24]. 

Overall, the above-mentioned research provides different perspectives on coach dismissals. However, coaches’ own perceptions and perspectives seem to be missing. In order to better support mental health and sustainability in the coaching profession, it is fundamental to consider the experience of the coaches. More specifically, there is limited research exploring elite football coaches’ own perceptions of getting fired. The aim of the current study was to get a deeper understanding of elite football coaches’ experiences of getting fired and how they made sense of that process.

## 2. Materials and Methods 

### 2.1. Design, Recruitment and Sample

Interpretative phenomenological analysis (IPA) is a methodological framework within qualitative psychology [34]. According to Smith [35] (p. 216), IPA is based on three theoretical cornerstones, with their particular hallmarks: phenomenology—exploring how events are experienced subjectively; double hermeneutic—an interpretative endeavor, as the researcher wants to understand how individuals make sense of an event; idiography—the researcher is concerned with understanding each of the participants in the study. Consequently, IPA is considered to be especially useful when exploring how individuals make sense of their experiences [36]. Further, the method helps the researcher focus on understanding, representing, and making sense of peoples’ way of thinking [37]. How the world appears to the individual is central to the method.

Based on the research question and IPA as a methodological framework, purposeful sampling of participants for the study was used [38,39]. In order to reach out only to those coaches that would respond to our purpose, it was decided to keep track of all dismissals among all head and assistant coaches within elite football (two highest divisions for males; highest division for females) in Norway during one season. During that season, eight coaches in total were fired (*n* = 8). All participants were invited to participate in the study, of which six accepted the invitation *(n* = 6). In this article, the data is anonymized, and the coaches are named Richard, Hannah, Thomas, George, Ian, and Luke. 

### 2.2. Data Collection

Data was collected using individual semi-structured interviews. The interview guide was relatively short and with few open questions as guided by IPA [39]. The intention of this interview guide was to help the participants talk freely about their personal experience of being fired. Further, probe questions were used throughout the interview (e.g., would you elaborate, could you please explain to me more what you meant by this in other words) to allow the participant and researcher to engage in a dialogue. The aim of the interview was to have an in-depth conversation about coaches’ own experience about the situation, where the coach was given space and flexibility in how he/she told his/her own story [34]. 

The first author conducted all the interviews. She has a pedagogical and psychological education and work experience from providing individual counseling within health care, which means developed interview and active listening skills [39]. Moreover, she has previously conducted interviews within the same elite population related to different research questions. This enhanced the ability to get closer to the topic, the coaches, and to the elite sport setting. Further, this population has massive exposure to media and routine interviews. It was therefore of importance to be clear with the participants and make them feel that the objective of this research was not to get sensational media coverage about the story of being fired but rather to get a deeper understanding and reflection about the process and sensemaking of the dismissal process from a coach’s perspective. This was deliberately discussed within the research theme before conducting the interviews and directly with the coaches before the interview started. Two interviews were conducted at the interviewer’s workplace, two interviews were conducted at the home of the coaches, and the final two interviews were conducted in the office of the coaches’ current employers. 

### 2.3. Data Analysis

All interviewees were transcribed verbatim, which resulted in 124 pages of single-spaced raw text. Further, the qualitative analysis software MAXQDA (VERBI Software, Berlin, Germany) was used for manually analyzing of the data guided by IPA. The analytic process within IPA is conducted through a process consisting of four steps [39]. First, the text was read multiple times to immerse in the data, where the first author also noted reflections about the process of the interview or thoughts of possible significance for interpretation. Secondly, the comprehensive notes were taken into account when aiming to transform the complete data set into emerging themes. This process was purely inductive when the researcher interpreted the overall tendencies and meaning making at a higher reflection level than the semantic data. The third step involved clustering smaller emerging themes with conceptual similarities to larger overarching themes. To guide this process, a constant interaction with the core components in the research question (“coaches own experience” and “how they made sense of process”) gave drive and direction to the work. During this stage, some emerging themes were dropped, both because they did not fit well with the emerging structure [39] and due to respect for third parties (see further elaboration under ethics). The final step of the analytic process relates to the presentation and writing up of the results. This will be done in the results section, where quotes from participants are presented to enable the reader to assess the interpretation done by the researchers and to retain the voice of the participants’ personal experience [39]. The first author led the analytic work, whereas the other researchers served as critical friends throughout the entire process [40]. Both the second and third authors have extensive experience working as psychological counselors within elite sport, and as such, their experience was of importance when interpreting the meaning of the participants’ experiences at a higher level. The discussion within the research team was important to ensure a higher quality of trustworthiness of the findings in the study [41]. 

### 2.4. Ethical Considerations

The study was approved by the Norwegian Social Data Services (project number 26524) and conducted in accordance with the Declaration of Helsinki. Before starting the interview, the participants and researcher together went through the informed consent form and all coaches signed this. By nature, elite environments within specific sports are both small and transparent. As Norway is a relatively small football nation, it could be argued that in this environment, it can be challenging to ensure anonymity. These coaches were all frequently interviewed, mentioned, and described in both local and national media. In particular, they have all been interviewed in relation to their dismissal from their football club. In this regard, it could be questioned whether we as researchers could ensure full anonymity, despite taking measures to protect them, including withholding information such as year of dismissal or combining gender and age. This was discussed with all coaches before they signed the consent form. They all agreed to participate under these terms. As the current research aims to better understand the coaches’ perspective, third parties such as their employers, colleagues, and athletes were not asked to participate. Therefore, some aspects and results of the study are not discussed due to respect and ethical considerations related to third parties. 

Due to the sensitive nature of the research topic, the interviewer anticipated the possibility of emotional reactions from the participants during the interview (e.g., sadness, anger, or feeling uncomfortable) [39]. Her experience as a counselor made her comfortable with the emotional reactions that occurred during the interviews. These situations were handled thoughtfully by giving the participant time, validation of emotions, and flexibility for participants to move forward in the interview as when he/she felt fit. 

## 3. Results

### 3.1. Descriptive Results

Participants’ descriptive data will be presented clustered to ensure anonymity. One coach was a woman, while five were men. Five worked as head coaches, while one worked as an assistant coach. Their age ranged from 39 to 58 years old (*M* = 45), while their professional experience ranged from 5 to 16 years (*M* = 9.5). All coaches had signed contracts, which included descriptions of the financial consequences of dismissal. Further, all coaches were in contact with and received legal support and advice from the National Football Coach Union after being fired. 

### 3.2. Main Results 

The analysis revealed five overarching themes related to how the football coaches’ experienced the process of getting fired and how they made sense of that process: “acceptance of having an insecure job”, “working for an unprofessional organization and management”, “micro-politics in the organization”, “unrealistic and changing performance expectation”, and “emotional responses”. 

#### 3.2.1. Theme 1: Acceptance of Having an Insecure Job 

All coaches accepted that they work in “a non-traditional occupation” regarding job security. Having been part of the elite environment for a large part of their adult lives, they acknowledged that their employment conditions were characterized by short term contracts, constant evaluation, a high degree of pressure, and a relatively high probability of dismissal. They all explicitly expressed that they accepted working in this culture and tradition. Acknowledging the insecurity associated with their employment, all expressed the importance of being able to handle these conditions in order to work at the elite level. This is how coach Richard expressed it with his words: “If you can’t handle it, you shouldn’t be a coach”.

#### 3.2.2. Theme 2: Working for an Unprofessional Organization and Management 

Despite being hired by elite football clubs, the coaches felt that their clubs were organized and managed unprofessionally. The clubs had full time administrative employees, including typical positions such as head of club, sports manager, marketing department, etc. Yet, an emerging finding was that even though the clubs were organized professionally, they were managed in an unprofessional way. On a general level, this was evidenced by lack of role clarity and role ambiguity between administrative employees and the board, which created uncertainty regarding responsibility and structure within the organization. The lack of role clarity, in addition to administrative staff that lacked understanding of elite sport, created challenges when, for instance, administrative staff tried to influence and interfere with decisions and that was within the responsibility of the coaching team. 

Furthermore, this theme was evident when coaches described how the process of dismissal was managed in a messy and inappropriate way. Several coaches felt that they were leading the process themselves by being the one in charge, and that it seemed like the responsible leaders did not own the dismissal process. Several coaches explained the dismissal meeting as both unprofessional and almost embarrassing, with leaders exhibiting avoidance behaviors, such as difficulty with eye contact and no hand shaking (i.e., expected at formal meetings).

“They had invited me to the meeting and we just sat there talking about the weather. I had to say—so you want to change the coach? And they said well...ehhh yeah...we have nothing to complain about, and you are a nice guy and all that, bla, bla, bla. It was like I had to comfort the vice-president” (Thomas). 

Another example of the unprofessional dismissal process from the coaches’ perspective was described as a total lack of process. Indeed, some of the coaches were called to a meeting with the leaders where the dismissal was conveyed with no advance warning. Hannah described the situation like this:

“Interviewer: So, there was no signal in advance that you might get fired?

Hannah: No. Nothing. Not at all, ehhh. Well, it had actually been in the newspaper that we were going to get fired, and on twitter. Whereas I had to respond vigorously to those spreading the rumors that it was not OK to spread false rumors. So, many in the football world certainly knew, but not us”.

#### 3.2.3. Theme 3: Micro-Politics in the Organization

The theme “micro-politics in the organization” stood out as a distinctive theme, yet it is related to the previous theme “unprofessional organization”. More specifically, this theme was characterized by “hidden agendas” in the dismissal process, as noted by all coaches. They had difficulty getting a clear picture of the reasons leading up to the dismissal decision. Consequently, all of them had questions about “whom, what, and why?”. The coaches talked about a range of stakeholders and agents with interest in the clubs that affected the process leading to dismissal. Keywords to describe what influenced the process were board members, media, money, personal beliefs, alliances, and power structures. George expressed it like this: “I got a phone call from the CEO, where he apparently had had conversations with key people, both formal and informal people, in the club. I had to prepare myself that they would start looking for a new coach”. Several of the coaches had an impression that the actual case for dismissal based on poor performance was “an excuse” to get the dismissal done and that there was something else going on in the background. Aligned with this, several of the coaches believed that the club, in times of poor performance, needed to fire the coach to show that they were being proactive: “They admitted that they were afraid of the surroundings (i.e., stakeholder, sponsors, media, and fans—authors note) and that they did not want give the impression that they were paralyzed” (George).

#### 3.2.4. Theme 4: Unrealistic and Changing Performance Expectations

Team performance was related to dismissal for all coaches. However, as previously described, coaches perceived that this could “not be the real reason” or was just part of the reason. Invariably, coaches found that poor performance was operationalized and understood differently between them and the different agents (those involved in the dismissal) in the sports club. Consequently, coaches’ perceptions of poor performance were not necessarily aligned with the evaluation of performance against either the objective results of the team, or the club’s expectations of results. More specifically, coaches perceived that they were fired due to performance that either was evaluated based on “unrealistic” or “changing” performance expectations. Unrealistic performance expectations were described by the coaches as a part of “the game”. For example, the elite clubs officially needed to have relatively high expectations for the current season to show that they were aiming high, to please and attract sponsors and to attract spectators to buy season tickets. The coaches felt that the CEO/club director (and their management) often set these goals without really collaborating with the coaches, whereas the coaches and team worked towards more realistic performance goals. Changing performance expectations was described by some of the coaches as a phenomenon that could occur if the team in the beginning of the season exceeded the expected level of performance, as illustrated by Ian’s comments: 

“At the start of the season we got really good results without playing well. And then, the expectations increased massively, and people started to talk about trophies. But then we had some losses, and poff, it was suddenly a crisis. Both supporters, media, the board and management, everything, it was like an extreme “downfall” even though this was what we could expect with the team we had at that moment”.

#### 3.2.5. Theme 5: Emotional Responses

Being fired from their position was described by all coaches as an undesirable outcome. When aiming to understand the coaches’ perceptions of the processes they had been through, their emotional responses stood out as an overarching theme. Not surprisingly when experiencing something undesirable, the coaches’ stories frequently identified negative emotional responses. These keywords describe what the coaches experienced: sadness, anger, frustration, disappointment, and sorrow. Several coaches felt that they were treated unfairly in the process and were being forced out of a job they originally were very passionate about. 

“I was disappointed and felt an extreme distrust in relation to the Board. Oh yeah, I know I have a fantastic salary, and I have a lot of responsibility, but I also know that I have put all my time, energy, and soul into this job (...). I thought it was difficult” (Ian).

However, negative emotional responses were not the only responses that stood out within this theme. Coaches also expressed a range of positive emotions experienced in the process. They described how they also felt a lot of support from both colleagues and players within the elite sport context, as well as from friends and acquaintances outside the sports context. Keywords several coaches used when describing this were grateful, appreciated, and supported. The following is a quote where Luke described the support he received: 

“I got like 70–80 bouquets of flowers on the door, it didn’t stop. Also, I had an older phone at the time, and I got like 3000 SMSs so the phone broke down and stopped working until two weeks afterwards. The support from people was overwhelming”.

## 4. Discussion

This is the first study to our knowledge that explores football coaches’ own experiences and their sensemaking of being fired. In various disciplines of psychology, it is evident that making sense of difficult experiences such as pain, severe stress, and suffering will have an impact on mental health. Central tenets within acceptance commitment therapy is to help people accept the things they cannot change, have courage to change the things they can, and have the wisdom to know the difference (adapted Serenity Prayer by Niebuhr) [42]. Consequently, our findings will be discussed based on coaches’ perception and sense-making of the process where they got fired. Overall, the findings indicate that all coaches participating in the study expressed awareness and acceptance of the risk of being fired. However, the coaches experienced a lack of transparency and clear feedback regarding the actual causes of dismissal. This caused frustration and some negative emotions based on a perception of unclear evaluation against poorly defined expectations and to some extent by anonymous stakeholders. This study focuses on the coach perspective in order to understand what it is like to have a job that comes with a high risk of losing it, and ultimately, what it is like getting dismissed. Altogether, the findings provide a deeper understanding of how stressors affect well-being and will be discussed within three main themes: beyond acceptance of job insecurity; who defines and what is acceptable performance; and coaches’ mental health in light of dismissals. 

### 4.1. Beyond Acceptance of Job Insecurity

It is well known that the consequences of job insecurity in organizations outside sport lead to negative outcomes at both individual and organizational levels [43,44,45]. Yet, it is reasonable to ask if the same consequences also apply to the population of elite football coaches represented in this study, since none of them expected a high degree of job security. In fact, they all agreed to work within an elite sport context with an awareness of the high risk of losing his/her job. In one way, they all seemed prepared for this potential outcome, and it was a part of their job contract. Acknowledging a realistic risk can be described as an adaptive coping strategy [46], with the ability to focus their energy on their primary work assignment—coaching—rather than spending too much energy on worrying about having an insecure job. However, it could be argued that accepting work under these conditions, with attitudes like “if you cannot handle it you should not be a coach”, may prolong a stereotypical image of what an elite coach is and what he/she should be able to handle. Keeping these stereotypical images alive might continue to limit the profession, as competent and ambitious coaches might choose to leave the occupation or decline to enter the elite sport context because they consider that it is not worth it or is simply not aligned with personal values [3,47]. This is particularly harmful if we want to promote greater diversity among those who take on jobs as elite coaches, such as for instance younger coaches and female coaches. Even though job insecurity seems to be somewhat accepted, it is also a pronounced stressor within the elite sport profession [3,12]. 

Beyond the finding of coaches’ acceptance of job insecurity, the results clearly indicated a non-acceptance of the way they were fired. Overall, the experience was that the dismissals were unprofessionally managed. In addition, there was a lack of clarity about how stakeholders were involved in the process towards the final decision about dismissal. Previous research from the managerial perspective has pointed to the power of stakeholders involved in dismissals of coaches, beyond the club directors. This could for instance be dissatisfied co-owners who hold a degree of power and expect their say in crucial matters [24]. These findings support the perceptions of the coaches in the present study who experienced the blurred micro-politics in play as an active ingredient within the process of losing their job. Micro-politic strategy has been defined as the “strategies by which individuals and groups in organizational contexts seek to use their resources of power and influence to further their interests” [48] (p. 88). Potrac and Jones have previously suggested that coaches job security not only depend on them being nice persons and winning games but also related to their gaining approval of relevant powerbrokers within the work context [49] (p. 229). Based on these results, it could be questioned whether sports organizations as employers have sufficient procedures and guidelines for dismissal of coaches and, further, to what extent these are transparent. In order for coaches to come to terms with how they were fired, this seems to be crucial. Notably, this is not the same as agreeing whether the dismissal was the right decision, as there will always (at least in most cases) be different perspectives. Rather, it could be argued that role clarity and role acceptance [34], both for the coaches and the managers in the club, are important factors that can minimize the risk of hidden agendas, chaos, and disorder (i.e., unprofessionalism) in the dismissal process. Moreover, previous research has also shown how role stressors (including role ambiguity, role conflict, and role overload) have detrimental effects on employers’ attitudes and strain responses at work [50]. This detrimental effect of role stressors is also underscored in light of the JD-R model [15], as role stressors are defined as a job hindrance [19]. So, role clarity seems to be equally relevant when coaches are doing their job, in the process of evaluation which potentially leads to dismissal and the process of dismissal itself. Based on the current findings, we argue that being fired will be more acceptable to coaches where there is a more transparent dismissal process, including role clarity throughout the process. From a mental health perspective, if coaches manage to accept not only job insecurity but also the process of dismissal, this is more likely to preserve mental health, based on research conducted with acceptance commitment therapy [51]. A more transparent process with clearer guidelines could also be argued to be more ethically sound, in relation to how an employer should treat their employees [52]. 

### 4.2. Who Defines and What Is Acceptable Performance? 

Related to the role of micro-politics in dismissals, a central finding of the study was the lack of agreement about what is considered acceptable performance. This question is already debated in the sports psychology literature, wherein scholars have argued a need to go beyond results to evaluate coach performance [53]. Yet, the results of the current study pointed to coaches’ perceptions of either being assessed on unrealistic or changing performance expectations. This raises the question of who has the power to define what an acceptable performance is, which seems crucial since previous research has reported that the performance aspect is at least the starting point of all football coach dismissals [24]. In addition, it gets even more blurred when some of the stakeholders raise their voices and have a say about performance expectations, and this is hidden in the micro-politics [49]. Further, if not all dismissals are based on objective results, should research on dismissals change focus from results to more appropriate and valid outcomes? Should football clubs develop a monitoring system with more relevant key performance indicators (KPIs) that actually assess the performance of the coach? In a prolonged discussion about acceptance, would such transparent KPIs add to the possible benefits of accepting the terms of dismissal for the coach as previously discussed. Some coaches in the present study were prepared for the dismissal process (having been evaluated along the way), whereas others were not. From the coaches’ perspective, they will always know that their performance is constantly under scrutiny, despite some variation on how, they relate to or cope with this. If a potential dismissal process leads to rumination, loss of energy, taking focus away from the coaching job itself, it will certainly put extra pressure on the coach, which will have a detrimental effect on their capacity and well-being. 

### 4.3. Coaches’ Mental Health in Light of Dismissals

Not surprisingly, when exploring coaches’ experiences of being fired, their emotional response emerged as a key theme. From a mental health perspective, for the coach population, being fired certainly was an unpleasant experience for all. None of them wanted this outcome, and they had all invested a lot of passion and energy in the job. Consequently, they all reported lack of energy, frustration, and a range of other negative emotions, which could all be considered symptoms of ill-being at work [7,8,54]. Further, the findings also indicated a decrease in occupational well-being [7], as their passion and love for the club and athletes was taken away. Even though they all knew this was “part of the game”, being dismissed also had a negative impact on their self-confidence as coaches, at least for the period close to the dismissal. In total, the coaches’ negative emotions and psychological responses (e.g., sadness, anger, frustration, disappointment, and sorrow) are aligned with the negative consequences reported in the general occupational literature regarding job insecurity, which includes low-self-esteem [55], symptoms of burnout and depression [56] and symptoms of anxiety [57]. The trend of dismissals in elite football is not decreasing [33], hence, there is a need to better understand, highlight, and acknowledge these negative consequences for the coach. A heightened awareness about the stressors related to both job insecurity and possible dismissal is essential in order to prevent occupational ill-being to some extent. This seems to be especially important in relation to chronic stressors that could be categorized as a job hindrance, compared to job challenges, as they have been shown to have more dramatic negative consequences on occupational health [17]. Consequently, we suggest that coaches should become better prepared during their coach education [58] and guided within their professional development, through for example mentoring programs [59], on how to handle both job insecurity and possible firings, considering the high frequency of such happenings in this particular job. 

Clearly this is not a study that promotes making the dismissal process a neutral or pleasant experience. Therefore, it was surprising to see positive emotions and psychological experiences emerging as findings from the coaches’ experience of being fired. The coaches were overwhelmed by the support they received in light of the firing. They also experienced good support from both individuals within and from outside the sporting community. Looking into research on life crises, aspects of positive emotions and appreciation of what you have are common outcomes [60]. This research also focuses on the preventive effects positive emotions might have when facing challenges in life [61]. Further, legal and process support from the National Coach Union also served as good support for several of the coaches. Within an applied perspective, Coach Unions can help coaches with practical, contractual, and financial issues in the dismissal process, which is invariably a difficult time. This could take some of the burden off the coach’s shoulders, who will often be providing emotional and practical support to his/her own family during the dismissal process in addition to coping with their own emotional responses. 

### 4.4. Limitations and Future Research

A limitation to the study is that the data for each coach was only collected through one interview with each coach. Even though the researcher conducting the interviews was familiar with the population of elite coaches, this was the only meeting between the coach and the researcher. Moreover, because of the sensitive topic of research, it is understandable if coaches do not openly discuss and share all their thoughts about the process. Further, this study also explores only the coaches’ perceptions of the process of being fired. In future studies, richer data collections [62] considering both several interviews with the same coaches aiming for fuller descriptions of their story and interviews with several stakeholders within the dismissal process (e.g., sport director, colleagues, athletes, media) could further illuminate the complexity of such incidents. 

Finally, there is also a need for some reflections about career transition, career termination, and whether this is voluntary or not. Within sports psychology researchers discuss termination of sports careers among athletes, and how this might be painful and harmful when not voluntary [63,64]. Few studies have explored career awareness, career planning, and career transitions among sports coaches [65,66], yet this seems to be an important resource to better manage some of the strain coaches’ experience as a result of having highly insecure jobs. Exploring how to better support coaches in post sport career planning and career transitions could be a good investment in coaches’ occupational health. 

## 5. Conclusions

It can be concluded that a key feature in high performance coaching is job insecurity and unrealistic performance expectations. Altogether, it can be argued, that making sense, acknowledging, and increasing awareness of these circumstances can help coaches to combat stressors in the profession. Nevertheless, no one is resilient to chronic stress, and mental health and sustainability in the profession may suffer. It is therefore suggested that stakeholders, sport and coach organizations, and employers should give this theme comprehensive attention and include specific subtopics such as job insecurity and monitoring key performance indicators and expectations. Resources allocated to this can be regarded as investment in coach well-being and sustainability, which ultimately would benefit all participants in the sporting community.

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
