# Peer review of "Elite Football Coaches Experiences and Sensemaking about Being Fired: An Interpretative Phenomenological Analysis"

_ijerph, 2020, doi:10.3390/ijerph17145196_

Round 1

Reviewer 1 Report

The paper reads very well and is clearly a topic which needs investigation from the coach’s perspective. The themes identified seem appropriate for discussion, however, there is some mention of potentially some secondary and primary themes. It may be beneficial to identify these primary themes as overarching or hierarchical.

The method section was very detailed providing a sound justification for the method used. The results section focused on previously identified themes and this could have been enhanced through identifying where these themes sit in relation to each other and the wider environment. The ethics section within this work was very well detailed and left no real further questions or issues around data collection.

The discussion section from line 321 looks at mental health perspectives, however these were not mentioned in the results section by participants neither were they asked specifically about this in the results section. Additionally, this has not been introduced as a factor within the introduction section of the work. It would be better if some indication of mental health was included at the start and then through the results section before being discussed here.

In section 4.2 from line 327, there needs to be further clarity around the link between the micro-politics in dismissal and the lack of agreement around acceptable performance. This is a really important point concerning acceptable performance and the paragraph raises some good discussion, however, the relationship to micro-politics in the dismissal process is not so clear.

From line 334 until 337, this section raises some good questions. However, it may well be more pertinent to bring together all of these solutions or thoughts under a section labelled implications. This would then be the take-home messages for coaches and managers.

Need to consider the earlier point around coaches mental health and the amalgamation with information from line 346. However, the issue of the lack of introduction and justification of this aspect of the coaching role still remains. Please see previous comments.

Line 351 to 357: whilst there is literature evidence to support these concepts, there was no data evidence reported from this study.

It is unclear whether the concept of making sense of the football coaches experience in dismissal has been achieved. Relevant information and data has been presented and this has been achieved very well. However, perhaps the title is misleading in terms of making sense of this situation.

Overall this article was a pleasure to read and is acknowledged to be dealing with some very difficult topics. It may be that my interpretation of some of these has been different to yours, the authors, and this may be related to a perceived mismatch between the title, the aims and some of the content. Thank you for the opportunity to read this.

Author Response

Response to Reviewer 1 Comments:

We thank the reviewer for the insightful and helpful comments to strengthen our manuscript. We have done our best to address the reviewer’s points in the revised version of it. We respond to the specific comments and suggestions below.

Comment regarding themes in result section

The themes in the result section emerged through the inductive IPA, so we are not sure if we understand the comment from the reviewer saying, “The results section focused on previously identified themes…”. We interpret that the reviewer suggests that the results / discussion would read better if we discussed how the five themes relate to each another. We have chosen not to do this explicitly. However, we have in the result section mentioned that: “The theme “micro-politics in the organization” stood out as a distinctive theme, yet it is related to the previous theme “unprofessional organization”.” (page 6, line 248-249.)

Comment regarding mental health

In the introduction, we have added a new paragraph that presents more thoroughly the concept of occupational health and the job demands-resources (JD-R) model (Bakker & Demerouti, 2007) as a theoretical framework to better understand how job characteristics influence both employees occupational health, addressing mental health, and performance. Further, we have drawn upon this new paragraph added to the introduction, when discussing our findings related to both mental health and characteristics in the work environment that influence coaches well-being.

The changes in introduction are displayed on page 1-2, from line 41 – 72.

The changes in the results section can be found on page 9, from line 396-398; page 9, from line 400 – 401; page 9, from line 407 – 410.

Comment regarding micro politics

More attention is given to the topics of micro-politics and role clarity/role stressors in the discussion. In addition, a number of references have been added.

These changes can be found on page 8, from line 347 -351; page 8, from line 358 – 362; page 9, from line 378-379.

Comment regard making new section of implications (previous lines 334 until 337)

We have strongly considered following the advice of the reviewer, bringing together all solutions / practical implication in one final paragraph. However, we have chosen to leave the suggestions of practical implication under each of the three subheadings in the discussion as we think the message that we try to convey reads best this way.

Comment regarding result of ill-being (previous lines 351 to 357)

We agree that the results of coaches’ symptoms of ill-being wasn’t explicitly repeated in the discussion, however their symptoms of illbeing were mentioned in the results. To make this clearer, we have included examples of coaches’ negative emotions and psychological responses in the discussion. Changes can be seen on page 9, line 396 – 402.

Comment regarding sensemaking

We agree with the reviewer that our message about coaches sensemaking almost disappeared throughout the manuscript. As this was the aim of the study, we have tried to make this more clear in several places in the manuscript. It has been made more explicit in the beginning of the results (p. 5, lines 206 – 207); in the beginning of the discussion (p. 7, lines 307-311); and in the conclusion (p. 10, line 446.

Reviewer 2 Report

This paper makes an important contribution to the field of elite sports research. As indicated, job insecurity has been a neglected area of enquiry in coaches. Research on the topic is important given the role that coaches play in managing their cohort teams, and the influence that coaches have on player wellbeing. The paper is clearly written and the methods are well described. There is detailed consideration of ethics regarding potential identification of participants. My comments offered are relatively minor, with the exception of better application of theory.

My main comment relates to a theoretical framework for interpretation of the results. Mention is made of job insecurity, adaptive coping, role clarity and role acceptance but a specific theoretical lens is not offered for interpretation of the findings. I would suggest the authors look the related literature for frameworks that have been used (perhaps role clarity – and for this to be introduced in the introduction). Such an addition may help bolster the interpretation of the results, and add greater depth and benefit future catalytic research in this area.

Minor comments.

Line 10 – “seem” should be “seems”

Line 14 – “were” should be “was”

Line 93 – Needs rewording “Therefore, it was kept track of all dismissals among all…”

Line 186 – Other subheadings are not italicised

Reporting of quotes – suggest a clear way to present the verbatim quotes so that they can be more easily distinguished e.g., indented and italicized.

Line 245 – “describe” should be “described”

Author Response

Response to Reviewer 2 Comments:

We thank the reviewer for the insightful and helpful comments to strengthen our manuscript. We have done our best to address the reviewer’s points in the revised version of it. We respond to the specific comments and suggestions below.

In the introduction, we have added a new paragraph that presents more thoroughly the concept of occupational health, and the job demands-resources (JD-R) model (Bakker & Demerouti, 2007) as a theoretical framework to better understand how job characteristics influence both employees occupational health, addressing mental health, and performance. The changes in introduction are displayed on page 1-2, from line 41 – 72.

We thank the reviewer for the minor comments regarding language, and all changes has been made accordingly.

Regarding the reviewer comment on reporting of quotes, unfortunately, the suggested changes can not be done due to the format (style) of the journal.

Reviewer 3 Report

I am not sure what we learn from this study other than football coaches in highly visible jobs are insecure about their employment.  Actually, their contracts contain clauses that enable them to get fired and still collect a sizable severance payment.  To make this manuscript interesting as a scholarly piece the problem needs to be placed in a larger context.  For example, it would be interesting to look at social and traditional media's influence on coaches being fired by owners. Perhaps the authors could research the extent to which fired coaches experience an identity crisis that propels them in some alternative career path.  Perhaps the authors could investigate how coaches manage the interactions with their players when they are fired.  The point is that just because little research on this topic has been done does not mean the research is worth doing.  Placing it in some larger context would be more interesting as a scholarly piece.

The second problem is that the paper offers no conceptual framework for guiding the inquiry.  Perhaps Social Identity Theory, Dissonance Theory, or Attribution Theory could be used to understand how coaches are making sense of their firing.  Then the study could be used to make a contribution to these theoretical frameworks.  Again, the research needs to have some kind of conceptual foundation to be relevant in a scholarly journal.   

Author Response

Response to Reviewer 3 Comments:

We thank the reviewer for the insightful, helpful and critical comments to strengthen our manuscript. We have done our best to address the reviewer’s points in the revised version of it. We respond to the specific comments and suggestions below.

Even though the coaches who get fired gets a sizable payment, all coaches within this population have to work with a constant stressor of job insecurity largely outside coaches’ control and hardly comparable to any other profession. We think this should be acknowledged, better understood, and prevented. A step in the right direction is to understand the stressor of job insecurity and potential dismissal in research from the coach perspective. It is our belief, that a constant risk of being fired, is an important and ethical matter that deserve scientific attention.

To the comment on placing the research within a larger context. We agree with the reviewer that it would be interesting to look at the challenge of dismissals / job insecurity also in a large picture. However, the data of the current study can only speak from the coaches’ perspective. In the paragraph about future research, we have suggested that several perspectives of the topic should be explored. Based on the reviewer comment, we have added media as one of several possible perspectives to examine in future research. This change can be found on page 10, line 435.

We agree with the reviewer that it is no automaticity in relation to a topic not being studied previously and that it should be. However, it is to our belief that the topic of dismissals for the coaches’ own perspective is needed and of importance. We have added a sentence in the paragraph to more explicitly explain why we find it interesting to explore. This sentence can be found on page 3, lines 108 – 110.

In the introduction, we have added a new paragraph that presents more thoroughly the concept of occupational health, and the job demands-resources (JD-R) model (Bakker & Demerouti, 2007) as a theoretical framework to better understand how job characteristics influence both employees occupational health and performance. The changes in introduction are displayed on page 1-2, from line 41 – 72.

Round 2

Reviewer 3 Report

This is a much improved paper with the addition of the conceptual grounding of the research.  I think it is ready for publication at this point.  Nice revision.